# Semicovariance Coefficient Analysis of Spike Proteins from SARS-CoV-2 and Other Coronaviruses for Viral Evolution and Characteristics Associated with Fatality

**DOI:** 10.3390/e23050512

**Published:** 2021-04-23

**Authors:** Jun Steed Huang, Jiamin Moran Huang, Wandong Zhang

**Affiliations:** 1School of Information Technology, Carleton University, Ottawa, ON K1S 5B6, Canada; jun.huang@carleton.ca; 2Department of Computer Science, Jiangsu University, Suqian 223800, China; moran@genieview.com; 3Human Health Therapeutics Research Centre, National Research Council of Canada, 1200 Montreal Road, Building M54, Ottawa, ON K1A 0R6, Canada; 4Faculty of Medicine, University of Ottawa, Ottawa, ON K1H 8M5, Canada

**Keywords:** fractional complex moment, SARS-CoV-2, coronaviruses, spike protein sequence, pearson correlation coefficient, semicovariance coefficient, positive-correlative and negative-correlative domains

## Abstract

Complex modeling has received significant attention in recent years and is increasingly used to explain statistical phenomena with increasing and decreasing fluctuations, such as the similarity or difference of spike protein charge patterns of coronaviruses. Different from the existing covariance or correlation coefficient methods in traditional integer dimension construction, this study proposes a simplified novel fractional dimension derivation with the exact Excel tool algorithm. It involves the fractional center moment extension to covariance, which results in a complex covariance coefficient that is better than the Pearson correlation coefficient, in the sense that the nonlinearity relationship can be further depicted. The spike protein sequences of coronaviruses were obtained from the GenBank and GISAID databases, including the coronaviruses from pangolin, bat, canine, swine (three variants), feline, tiger, SARS-CoV-1, MERS, and SARS-CoV-2 (including the strains from Wuhan, Beijing, New York, German, and the UK variant B.1.1.7) which were used as the representative examples in this study. By examining the values above and below the average/mean based on the positive and negative charge patterns of the amino acid residues of the spike proteins from coronaviruses, the proposed algorithm provides deep insights into the nonlinear evolving trends of spike proteins for understanding the viral evolution and identifying the protein characteristics associated with viral fatality. The calculation results demonstrate that the complex covariance coefficient analyzed by this algorithm is capable of distinguishing the subtle nonlinear differences in the spike protein charge patterns with reference to Wuhan strain SARS-CoV-2, which the Pearson correlation coefficient may overlook. Our analysis reveals the unique convergent (positive correlative) to divergent (negative correlative) domain center positions of each virus. The convergent or conserved region may be critical to the viral stability or viability; while the divergent region is highly variable between coronaviruses, suggesting high frequency of mutations in this region. The analyses show that the conserved center region of SARS-CoV-1 spike protein is located at amino acid residues 900, but shifted to the amino acid residues 700 in MERS spike protein, and then to amino acid residues 600 in SARS-COV-2 spike protein, indicating the evolution of the coronaviruses. Interestingly, the conserved center region of the spike protein in SARS-COV-2 variant B.1.1.7 shifted back to amino acid residues 700, suggesting this variant is more virulent than the original SARS-COV-2 strain. Another important characteristic our study reveals is that the distance between the divergent mean and the maximal divergent point in each of the viruses (MERS > SARS-CoV-1 > SARS-CoV-2) is proportional to viral fatality rate. This algorithm may help to understand and analyze the evolving trends and critical characteristics of SARS-COV-2 variants, other coronaviral proteins and viruses.

## 1. Introduction

Complex algorithms are used to analyze real-world implementations, i.e., it comes as the trusted analytic solution, but typically tends to have challenges in software implementation complexity requiring simpler software solution by using complex theory. Complex algorithms have received significant attention in recent years and are increasingly used to solve real-world problems, among which are the combination of two or more algorithms involving numerical algorithms, analytic calculation [1], and other computational techniques such as artificial intelligence [2,3,4], gene analysis systems [3] or gene simulation [4].

The Fractal DNA hypothesis (FDH) was first introduced by at least three groups independently in 1992 [5]. The arithmetic data from the DNA sequences by counting the number of intervening bases from a specific base (A) to the next one, etc. (inter-event data), are characterized by a dynamical process whereby long-range (fractal) correlations are observed. Being different from the traditional DNA hypothesis, RNA and protein analysis is based on fragment length between the domains with electrical charges. Especially, it emphasizes the influences on the behaviors of charges caused by the difference of information reception and lengths of expression or neighbor status observing the existence of fractal structure in stable DNA [6]. Some work around FDH on RNA has been reported recently [7] where the genetic sequences were converted to binary numbers, purines converted to −1 and pyrimidines converted +1. The dimension order was found to be SARS-CoV-1 > SARS-CoV-2 > MERS, which differs from the time evolution order. Thus, we wish to further examine the similar electrical charge specific (+1 for positive amino acid, −1 for negative counterpart, 0 for neutral one, Inter-event data) relationship among the coronaviruses. The SARS-CoV-2 virus is among the longest positive single-stranded RNA virus, and its protein folding/tertiary structure and functions are closely related to the charges of the amino acid residues. It is therefore important to examine the charging structure/patterns or nonlinear correlation patterns of the spike protein of SARS-CoV-2 as compared to the spike proteins from other coronaviruses to understand viral evolution and the characteristics of the spike proteins associated with viral virulence and fatality.

The FDH does not distinguish the values above or below the mean (average) of the DNA fragment length between the gene signatures. Our algorithm used in this study focuses on distinguishing the values above and below the mean (average) to calculate the semicovariance coefficient of the spike protein sequences from coronaviruses, including SARS-CoV2. The higher value above the mean indicates higher similarity and increased evolutionary conservation, while the lower value below the mean indicates more dissimilarity and increased variations/mutations. Analysis with our algorithm can be carried out rapidly by running the Microsoft Excel sheet tool. In our study, the traditional Pearson correlation coefficient for the spike protein sequences of coronaviruses [8] is also calculated for comparison [9]. By imaging the charge similarity covariance as a weight (gravity or Coulomb force) on the rod of the axis, the weight center of semicovariance coefficient is calculated to examine the evolving weight center (both convergent (positive correlative) center/region and divergent (negative correlative) center/region) shifting pattern of the spike protein sequences of coronaviruses [10] and identify spike proteins’ characteristics associated with viral virulence and fatality.

## 2. Materials and Methods 

### 2.1. Coronaviruses and Spike Protein Sequences

The coronavirus spike protein sequences used in this study were obtained from the NCBI GenBank and the GISAID databases, including SARS-CoV-2 (the sequences of the virus strains isolated in Wuhan, Beijing Xinfadi wholesale market, Germany, New York, UK (Wales), and New York Zoo tiger), SARS-CoV-1, Middle East respiratory syndrome (MERS), bat coronavirus (RaTG13), pangolin coronavirus, feline coronavirus, canine coronavirus, and swine coronaviruses (Swine Transmissible gastroenteritis virus (Swine-stomach), swine enteric coronavirus (Swine-Ent), and porcine respiratory coronavirus (Swine-Res)). The sequence IDs from the GenBank and GISAID databases are listed in Table 1.

### 2.2. Hypo, Hyper or Gauss Variances and Covariance

The complex parameter is a measure of fluctuation-term memory time series. It relates to the autocorrelation time series, and the derivatives of Laplace transformation (frequency spectrum) time series or the momentum generation function at the origin [11]. Studies involving the complex parameter were originally developed by Jerome Cardan (1501–1576) for solving algebra equations.

In order to calculate the fractional version of the center momentum, we need to first generalize the binomial formula from integer domain to real domain, as described previously [12]. Here, we only show the formula:(1)$$\mu_{k}=E{\left(\xi -E\right)}^{k}=E\left(\overset{\infty }{\underset{i=0}{\sum }}\left(\begin{array}{c} k\\ i \end{array}\right){\left(-1\right)}^{i}\xi^{k-i}{\left(E\xi \right)}^{i}\right)\;\;\;$$
(2)$$\;\;\;\;=\overset{\infty }{\underset{i=0}{\sum }}\left(\begin{array}{c} k\\ i \end{array}\right){\left(-1\right)}^{i}E\left(\xi^{i-k}\right)E^{i}\xi \;\;\;\;\;\;\;\left(1<k<3\right)\;\;\;\;$$
when *k* = 2, it is Gauss variance; *k* < 2 is hypo version; *k* > 2 is hyper version. Factorial of fractional *k* is calculated by Gamma function. From which we can also have the covariance counterpart:(3)$$\nu_{2k}=E{\left(\xi -E\xi \right)}^{k}{\left(\eta -E\eta \right)}^{k}\;\;\;\;\;$$
(4)$$=E(\overset{\infty }{\underset{i=1}{\sum }}\left(\begin{array}{c} k\\ i \end{array}\right){\left(-1\right)}^{i}\xi^{k-i}{\left(E\xi \right)}^{i}\overset{\infty }{\underset{i=1}{\sum }}\left(\begin{array}{c} k\\ i \end{array}\right){\left(-1\right)}^{i}\eta^{k-i}{\left(E\eta \right)}^{i})\;\;\;\;\;\;$$
(5)$$\rho =\frac{\nu \left(real\right)+i\times \nu \left(img\right)}{\mu_{\xi }\times \mu_{\eta }}\;\;\;\;\;$$
where $\nu \left(real\right)$ is the positive covariance, $\nu \left(img\right)$ is the negative covariance. Define the basic rectified linear unit as *ReLU*(*X*) = max(0,*X*), we can have the simplified semicovariance related back to the traditional Pearson correlation as below:(6)$$\begin{array}{l} Pearson\left(x,y\right)=\frac{E\left(\left(X-EX\right)\left(Y-EY\right)\right)}{\mu_{\xi }\times \mu_{\eta }}=\frac{E\left(ReLU\left(\left(X-EX\right)\left(Y-EY\right)\right)\right)}{\mu_{\xi }\times \mu_{\eta }}-\frac{E\left(ReLU\left(-\left(X-EX\right)\left(Y-EY\right)\right)\right)}{\mu_{\xi }\times \mu_{\eta }}\\ \;=\nu \left(real\right)/(\mu_{\xi }\times \mu_{\eta })-\nu \left(img\right)/\left(\mu_{\xi }\times \mu_{\eta }\right) \end{array}$$

In sum, the Pearson coefficient is the difference between the positive part (the first and the third quadrants) and the negative part (the second and the fourth quadrants) which are the proposed semicovariance coefficients. The product of two differences (the two values above the mean and the two values below the mean) on the same side of the mean value will be the real part, or the convergent part so that we can call it positive correlation covariance coefficient. The product that is on the opposite side of the mean will be the imaginary part, or the divergent part so that we can call it negative correlation covariance coefficient [13].

## 3. Results

### 3.1. Excel Calculations of Semicovariance for Spike Proteins from SARS-CoV-2 and Other Coronaviruses 

To compare and prove the usefulness of the simplified complex variances, we compare the correlation of SARS-CoV-2 viral spike protein sequence with other coronavirus spike protein sequences [14]. Since Excel is not capable of handling the imaginary number, we simplify the calculation with integer power, but separate the positive and negative covariance signs [15]. As coronaviral spike proteins have different electrical charge levels [16], we normalize the covariance by the variance, respectively just as the Pearson calculation does [17]. We calculated the sequences of the spike proteins [18] and plotted the curve starting from the N-terminus to the C-terminus. By using the moving window (a typical peptide) of 16 neighborhood amino acid residues [19], we calculated the covariance and average/mean over the same period of sequences to make the curve visually smooth for easier comparisons [20]. We selected the window size 16 to maximize the information we can extract from the charge even data, as it was the boundary between oligopeptide and polypeptide for the number of the amino acids.

Here are the steps for Excel sheet calculations:

(1)Collect data from GenBank and GISAID, align up the spike position, and note the similarity and difference. (2)Convert the aligned sequence into charge data, i.e., positive is +1; negative is −1; and the rest is 0.(3)Calculate the Pearson coefficient between each trace and the baseline trace (Wuhan strain).(4)Find the maximum Pearson value and the corresponding amount of shifting.(5)Line up the trace by removing the extra shift points or padding with the zeros.(6)Calculate the mean charge value with moving window size of 16. (7)Calculate the difference to the mean value for each point in each trace.(8)Calculate the semicovariance by separating the positive and negative product values.(9)The positive part is real value corresponding to convergent data for classifying the series. (10)The negative part is imaginary value corresponding to divergent data for fatality reasoning.(11)Smooth out the curve by moving average with window size 16 for better visual display.(12)Summarize the second page (semi) and the third page (Pearson) in the first page of the sheet.

We defined the conserved (and diverged) centers or regions of the spike proteins for different coronaviruses. For example, the conserved centers of SARS-CoV-1, pangolin and bat coronaviruses are located at the amino acid residues 905, 906 and 904, respectively. These viral spike proteins are set as the 900 series (Table 1). The conserved center is defined as the weight center of the spike protein sequence at which the charge pattern before and after those points is the same. The conserved center for SARS-CoV-2 is at the amino acid residue 658, and the conserved center for feline coronavirus is at the amino acid residue 683 (Table 1). Thus, the SARS-CoV-2 and feline coronaviral spike proteins are defined as the 600 series. The conserved centers for MERS and three swine coronaviruses range from amino acid residues 727 to 784, hence they are defined as the 700 series. Therefore, as the latest SARS-CoV-2 variant, B.1.1.7 from the UK, has its conserved center at 702, this suggests that the charge pattern of the variant B.1.1.7 spike protein has evolved to 700 series from 600 series and that the variant B.1.1.7 may be more virulent or deadly than the original SARS-COV-2 strain. 

Figure 1, Figure 2, Figure 3, Figure 4, Figure 5 and Figure 6 are the calculation results from our algorithm of semicovariance coefficient for spike protein of the Wuhan SARS-CoV-2 strain in comparison with spike proteins of other coronaviruses listed in Table 1. Figure 7, Figure 8, Figure 9 and Figure 10 are the corresponding scatter graphs. Figure 11 illustrates the center and maximum positions listed in Table 2. 

The spike protein sequences were analyzed with index order from animal coronaviruses (pangolin, bat, canine, swine, feline, and tiger) and human coronaviruses (SARS-CoV-1, MERS, and SARS-CoV-2) [21]. Figure 1 presents the calculation results of the spike proteins for human coronaviruses including 600 (SARS-CoV-2)//700 (MERS)//900 (SARS-CoV-1) series of spike proteins semicovariance selected based on Table 1. Figure 2 presents the analysis for coronaviruses whose conserved center is located from the spike protein amino acid residues 900 to 999 (900 series) as described above. Figure 3 is for 800 and 700 series. Figure 4 is for 700 and 600 series. Figure 5 is for 600 series. Figure 6 is for 600 and 700 series.

### 3.2. Pearson and Semicovariance Coefficient Analysis of Spike Proteins from Coronaviruses

Table 1 compares the Pearson correlation coefficient analysis with semicovariance coefficient analysis for coronaviral spike proteins. From Table 1, it shows that the Pearson correlation coefficient only reflects the variation after the cancellation of up and down correlation [22]; however, our semicovariance coefficient reflects the direction of the variations before the cancellation of correlation [23]. 

To visually examine the nonlinear correlation relationship between Wuhan strain SARS-CoV-2 and the rest of the coronaviruses, we further plotted scatter graphs where the *X*-axis is the charge variation over the 16 neighborhood amino average of the Wuhan strain of SARS-CoV-2, and the *Y*-axis is the charge variation over the 16 neighborhood amino average of the respective virus (Figure 7, Figure 8, Figure 9 and Figure 10). It can be seen that some of them have nonlinear relationships. The Pearson correlation may not be good enough to depict all of them. The second quadrant and fourth quadrant represent the strong mutation part where the charge is reversed, the first quadrant and the third quadrant are the weak mutation part where the charge is not reversed (Figure 7, Figure 8, Figure 9 and Figure 10).

It can be seen that there are a combination of linear and nonlinear relationships in Figure 7A,B; while Figure 7C shows a linear relationship. Figure 8A–C show a linear relationship for the scatter patterns of the spike proteins from the viral strains isolated in German, New York and Beijing relative to that of the Wuhan strain, indicating a high similarity. Figure 7C is identical to those of Figure 8A–C, suggesting that the same strain of the SARS-CoV-2 was transmitted from human to the New York Zoo tiger. All of these viral strains carry the D614G mutation in the spike protein. However, Figure 8D shows the UK variant B.1.1.7 vs. the Wuhan strain and there is a combination of linear and nonlinear relationships between them, indicating that the mutations in B.1.1.7 result in amino acid changes with opposite charges as compared to the Wuhan strain. Figure 9A shows a combination of linear and nonlinear relationships between SARS-CoV-1 and Wuhan strain SARS-CoV-2 spike proteins; while Figure 9B shows a nonlinear relationship between MERS and Wuhan strain SARS-CoV-2 spike proteins, indicating a strong dissimilarity between them. Figure 10 shows nonlinear relationships between those animal viral spike proteins and the Wuhan strain SARS-CoV-2 spike protein, indicating a strong dissimilarity between them. However, the local piecewise similarity island pattern is still clearly seen. That means they are still related somehow. 

Table 2 uses the spike protein sequence from Wuhan SARS-CoV-2 strain as a reference to compare with SARS-CoV-2 strain isolated in Beijing (carrying D614G mutation), SARS-CoV-1, and MERS. It incorporates fatality rates to identify critical amino acid regions associated with mortality. As compared to SARS-CoV-2, there are 74 amino acid residues in MERS spike protein sequence that are critical to MERS-associated fatality, and there are 18 amino acid residues that are associated with SARS-CoV-1 fatality. There are only nine amino acid residues in the Beijing strain viral spike protein that are different from Wuhan strain SARS-CoV-2 spike protein. The correlation coefficient of the analysis for these critical amino acids in the spike proteins associated with fatality is R = 0.9981 among the three coronaviruses infecting humans. The similar calculation for the ratio of Mutation Coulomb force center to maximum Coulomb force point leads to R = 0.9958. Divergent Coulomb intensity dictates the fatality. The diverged center (imaginary part) is illustrated in Figure 11; the same definition applies to the conserved (real) part. The definition of the maximum and the center of the divergence is illustrated in the same figure.

## 4. Discussion 

This study presents the construction of a complex covariance for the fractional analysis of coronavirus spike proteins by using a fractional moment based simple algorithm coded in an Excel Sheet. The analysis with our novel complex model reveals an additional performance index over the traditional real model, such as the Pearson correlation coefficient. Our model compares the traditional Pearson calculation of the integer dimension against the fractional dimension. The complex calculation shows the differences among viral spike proteins, which the traditional covariance definition and calculation may overlook. Our study reveals the unique convergent (positive correlative) to divergent (negative correlative) centers of each virus and the distance/length between the positive- and negative-correlative centers/regions (Table 1). Interestingly, we found that the distance between divergent center (mean) and the maximal divergent point is associated with viral fatality. As compared to the SARS-CoV-2 strain isolated in Wuhan, the distance between the divergent center (mean) and the maximal divergent point is located at the amino acid residue 614 in the SARS-CoV-2 viral strains isolated in Beijing, Germany, New York and New York Zoo tiger. This suggests those viruses are essentially the same except at amino acid 614 (D614G mutation, aspartate (D) to glycine (G)) also reported in the literature [24,25]. While the distance between the divergent center (mean) and the maximal divergent point in the spike protein of SARS-CoV-1 is from the amino acid residues 309 to 338 (Table 1), the distance between the divergent center (mean) and the maximal divergent point in the spike protein of MERS is from the amino acid residue 214 to 698 as compared to the spike protein of Wuhan strain SARS-CoV-2 (Table 1). It is evident that the fatality rate caused by the virus is highly related to the distance between the divergent center (mean Coulomb force) and the maximal divergent Coulomb (force) point (Table 2). The longer the distance, the more mutations (Coulomb force) and the more deadly and virulent the virus is. This region of the MERS spike protein occurs with a high frequency of variations as compared with the spike proteins from other coronaviruses and may be responsible for its high fatality. 

From Table 1, it is shown that our complex coefficient reveals more dependency and trends of each protein sequence’s evolution [26]. In the past, the viral spike protein’s conserved center evolved from the amino acid residue 900 in SARS-CoV-1 down to 600 in SARS-CoV-2. The conserved region or convergent center may be critical to the viral stability or viability. This conserved center/region of the viral spike protein has been shifted from SARS-CoV-1 at the amino acid residue 900 to amino acid residue 700 in MERS spike protein, and then shifted to amino acid residue 600 in SARS-CoV-2. The charge pattern of the SARS-CoV-1 spike protein sequence around 900 is similar to that of the MERS spike protein around 700, and similar to that of the SARS-CoV-2 around amino acid residue 600. Interestingly, the convergent center of the UK variant B.1.1.7 is shifted from 600 in SARS-CoV-2 strains (Wuhan, German, New York, and Beijing strains) to 700 (Table 1). The convergent center of the UK variant B.1.1.7 spike protein in 700 is similar to those of MERS and swine coronaviral spike proteins (Table 1), which may indicate greater lethality as compared with the SARS-CoV2 strains isolated in Wuhan, German, New York, and Beijing. Our analysis suggests that the conserved center/region may be essential for the biology, viability, and evolution of the coronaviruses. This conserved center/region may shift to a new location in new SARS-COV-2 variants or other novel coronaviruses.

## 5. Conclusions

In this study, we have analyzed spike protein charge patterns of coronaviruses by using our algorithm of semicovariance (nonlinear) coefficient as compared to the Pearson (linear) correlation coefficient, (based on the original semivariance principle [27] for risk analysis initiated by 1990 Nobel Prize winner Harry M. Markowitz). The analysis reveals an additional performance index over the Pearson analysis, such as both positive- and negative-correlative centers/regions in the spike proteins. The study reveals that the distance between the divergent center (mean) and the maximal divergent point is associated with viral fatality. The longer the distance is, the more variations/mutations are, and the more deadly the virus is. The correlation coefficient analysis also identifies the critical amino acids in the spike proteins associated with fatality among the three coronaviruses infecting humans. Our study suggests that the conserved center/region of spike proteins identified by the analysis is essential for the biology and viability of the viruses and that the shifting of this region involves the evolution of the viruses. The conserved center/region of the UK variant B.1.1.7 has shifted to the MERS category of viruses, indicating more virulence of this variant. In addition, the analysis provides an in-depth understanding for the nonlinear viral evolution pattern and identifies the protein Coulomb force characteristics which may be associated with viral fatality.

It is envisioned that this complex number model is a good alternative for the covariance analysis of coronaviral spike proteins. This type of analysis may go beyond asymmetrical fluctuations to help in developing high dimensional Fractal theory. However, the simplified calculation is easier for practical analysis and applications. The simplified Excel sheet calculation is very easy to use, accurate and forward compatible with traditional Pearson model and calculations. The example code is available from the Excel file on the GitHub server (https://github.com/steedhuang/COVID-19-gene-convertor) accessed on 21 April 2021. Our future work will look into other viral proteins with the same methodology for viral evolution and the Coulomb characteristics that are associated with viral fatality. More attention will be paid to the relationship between positive charges and infectivity. Additionally, we will use an unsupervised machine learning algorithm such as k-means classification to find the optimum moving window size (for peptide) and to predict the next mutation spot (region), and use a supervised machine learning algorithm such as Recurrent Neural Network (RNN) to find the potential upcoming virus variant(s) emerging time [28]. These analyses may allow the vaccine and antibody therapies to be prepared ahead of time before the new variants appear.

## Figures and Tables

**Figure 1 entropy-23-00512-f001:**
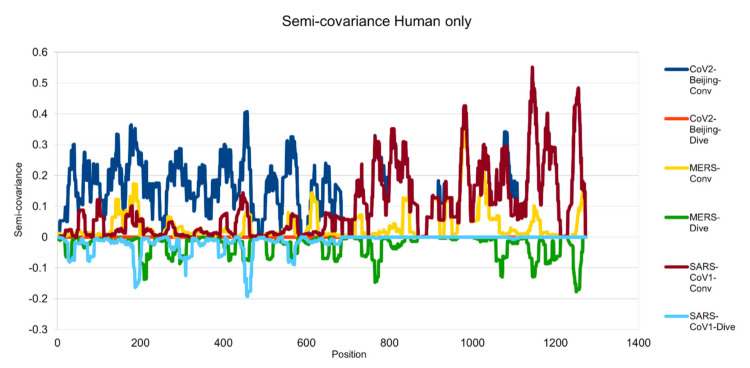
Semicovariance coefficient among the spike proteins from human coronaviruses SARS-CoV-2, SARS-CoV-1 and MERS. The spike protein sequence of Wuhan SARS-CoV-2 strain is used as a reference for comparison with the spike proteins of SARS-CoV-2 viral strain isolated from Beijing Xinfadi wholesale market (carrying D614G mutation), SARS-CoV-1, and MERS. Mathematically speaking, the diagram/curve above the *X*-axis is the positive correlation (convergent or conv in the figure). The higher the value, the greater the similarity of charge patterns between the compared viral spike proteins. SARS-CoV2 strain isolated in Beijing Xinfadi wholesale market is the same as the SARS-CoV-2 strain isolated in Wuhan throughout the entire sequence except D614G mutation. SARS-CoV-1 shows a similar pattern with SARS-CoV-2 after amino acid residue 700; while MERS shows a similar pattern with SARS-CoV-2 only around the amino acid sequence 1000. The second similar region of the MERS spike protein sequence with SARS-CoV-2 lies around amino acid residue 200. The diagram/curve below the *X*-axis is the negative correlation (divergent or dive in the figure). The lower the value, the more oppositely charged, thus the greater the disdissimilarity between the compared viruses. MERS has more opposite charges around amino acid position 200, 800 and 1200 as compared to SARS-CoV-2; while SARS-CoV-1 has a few opposite charges around amino acid position 200 and 450 as compared to SARS-CoV-2, indicating more similarity between the spike proteins from SARS-CoV-1 and SARS-CoV-2 but not between SARS-CoV-2 and MERS.

**Figure 2 entropy-23-00512-f002:**
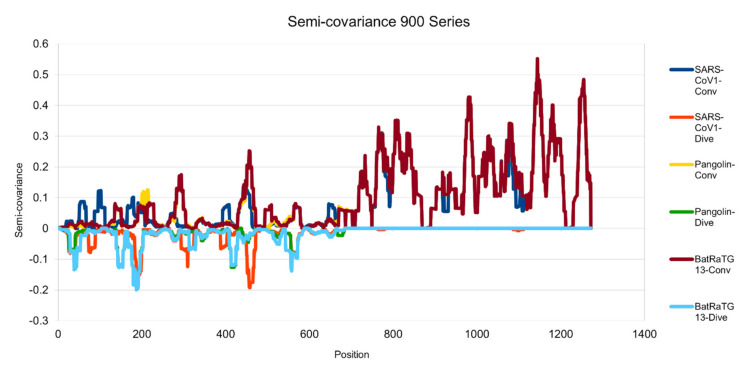
Semicovariance coefficient of SARS-CoV-2 spike protein with the spike proteins from SARS-CoV-1, pangolin and bat coronaviruses (900 series). The diagram/curve above the *X*-axis is positive correlation (convergent or conv in the figure). The higher the value, the greater the similarity of the charge patterns between the compared viruses. The spike protein of SARS-CoV-1 is similar to the spike protein of Wuhan SARS-CoV-2 from amino acid residues 700 onwards, as well as the spike proteins from pangolin and bat coronavirus. The spike proteins of SARS-CoV-1, bat and pangolin coronaviruses overlap each other more after amino acid residues 700. The diagram/curve below the *X*-axis is negative correlation (divergent or dive in the figure). The lower the value, the more oppositely charged, thus the more dissimilarity between the compared viruses. The spike proteins of the pangolin and the bat peak around position 200, suggesting that the charge pattern is not similar at this region or the variations/mutations occurred more at this region between SARS-CoV-2 and the pangolin/bat coronaviruses. Similarly, SARS-CoV-1 peaks around 200 and 450 amino acid residue positions, suggesting that the charge patterns are different between SARS-CoV-1 and SARS-CoV-2 at this region or the mutations have made this region different between the two viruses.

**Figure 3 entropy-23-00512-f003:**
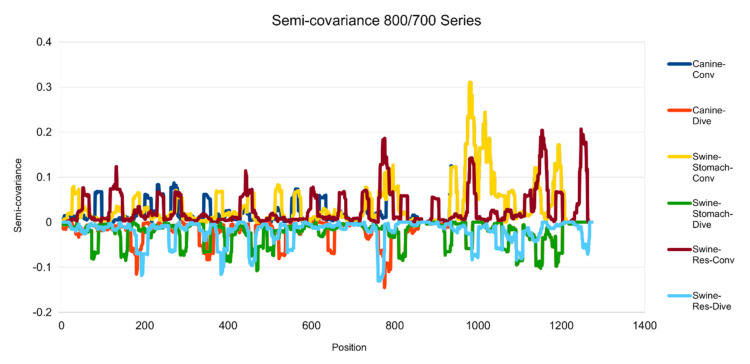
Semicovariance coefficient of SARS-CoV-2 spike protein with the spike proteins from MERS, canine and swine coronaviruses (700/800 series). The diagram/curve above the *X*-axis is the positive correlation (convergent or conv in the figure) between the compared viral spike proteins. The higher the value, the greater the similarity of charge patterns among the compared viruses. Canine and swine transmissible gastroenteritis virus (Swine-Stomach) are similar to Wuhan SARS-CoV-2 strain from the amino acid residue 1000 onwards, as well as porcine respiratory coronavirus (Swine-Res), but with less similarity. The diagram/curve below the *X*-axis is the negative correlation (divergent or dive in the figure). The lower the value, the more oppositely charged and the greater the dissimilarity between the compared viruses are. The spike proteins from canine, swine transmissible gastroenteritis virus (Swine-Stomach) and Swine-Res coronaviruses have opposite charge patterns at different positions. The spike proteins from canine and Swine-Res coronaviruses peak around 800, and the one from Swine-Stomach coronavirus peaks around amino acid residue positions 450 and 1150.

**Figure 4 entropy-23-00512-f004:**
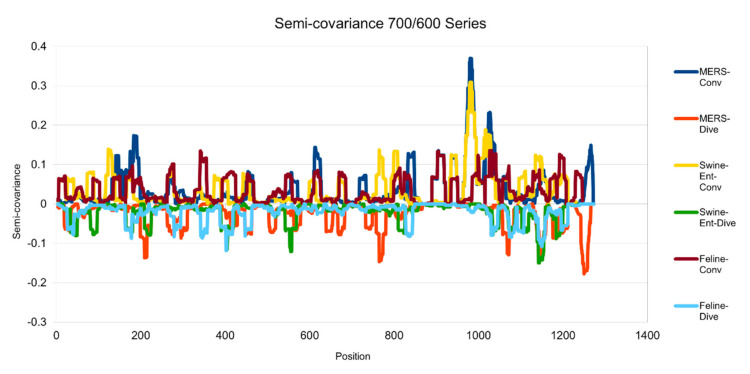
Semicovariance coefficient of SARS-CoV-2 spike protein with the spike proteins from MERS, swine and feline coronaviruses (700/600 series). The diagram/curve above the *X*-axis is the positive correlation (convergent or conv in the figure). The higher the value, the greater the similarity of charge patterns between the compared viruses. The spike proteins from MERS and Swine-Ent coronaviruses are similar to Wuhan SARS-CoV-2 around amino acid residues 1000; while the spike protein from feline coronavirus is not similar to Wuhan SARS-CoV-2. The diagram/curve below the *X*-axis is the negative correlation (divergent or dive in the figure). The lower the value, the more oppositely charged, the greater the dissimilarity between the compared viruses. The spike proteins from MERS, Swine-Ent and feline coronaviruses have opposite charge patterns at different positions. The spike proteins of feline and Swine-Ent coronaviruses peak around 400 and the spike protein of MERS peaks around amino acid residue positions 750 and 1250.

**Figure 5 entropy-23-00512-f005:**
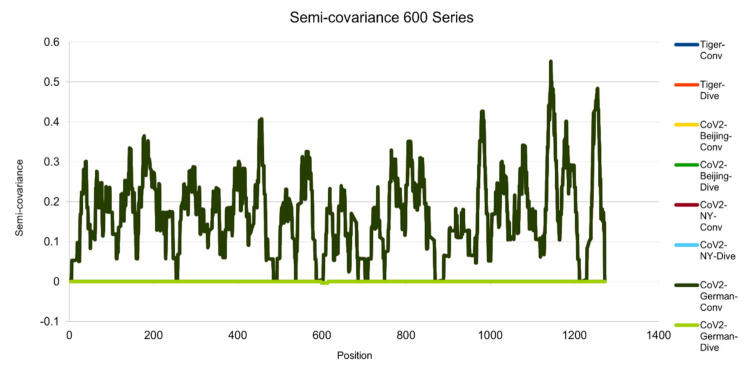
Semicovariance coefficient of Wuhan SARS-CoV-2 spike protein with the spike proteins of SARS-CoV-2 isolated in Beijing, New York, Germany, and New York Zoo tiger (600 series). The diagram/curve above the *X*-axis is the positive correlation (convergent or conv in the figure). The higher the value, the greater the similarity of charge patterns between the compared viruses. SARS-CoV-2 isolated in Beijing Xinfadi wholesale market, New York, Germany and the New York zoo tiger overlap and are almost identical to Wuhan SARS-CoV-2. The diagram/curve below the *X*-axis is the negative correlation (divergent or dive in the figure). The lower the value, the more oppositely charged and the greater the dissimilarity between the compared viruses are. SARS-CoV-2 spike proteins from Beijing, New York, Germany and the New York zoo tiger carry D614G mutation and have the only opposite charge at amino acid residue 614 position (D614G mutation as reported in the literature).

**Figure 6 entropy-23-00512-f006:**
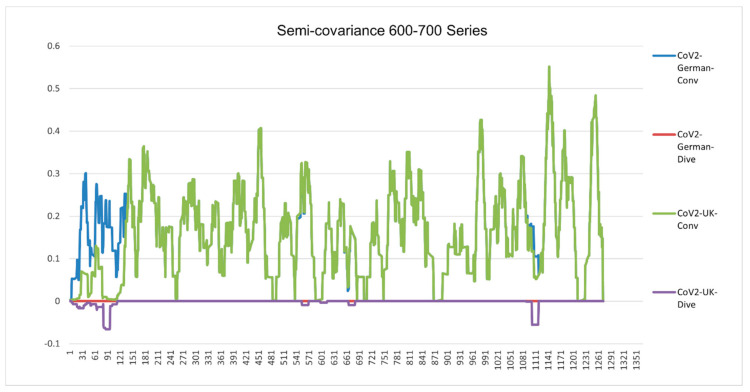
Semicovariance coefficient of Wuhan SARS-CoV-2 spike protein with the spike proteins of SARS-CoV-2 isolated in UK (variant B.1.1.7) (700 series) and Germany (600 series). The diagram/curve above the *X*-axis is the positive correlation (convergent or conv in the figure). The higher the value, the greater the similarity of charge patterns between the compared viruses. SARS-CoV-2 variants isolated in the UK (Wales) (B.1.1.7) and Germany are almost identical to Wuhan SARS-CoV-2, except the beginning part of B.1.1.7 is mutated back to SARS-COV-1. The diagram/curve below the *X*-axis is the negative correlation (divergent or dive in the figure). The lower the value, the more oppositely charged and the greater the dissimilarity between the compared viruses are. The UK variant B.1.1.7 has not only the opposite charge around 600 amino acid residue position (D614G mutation as reported in the literature) but also at the old one around 100 (SARS-CoV-1) and at a new position (1000). The mutation sites occur towards both sides of the 600 series. It flips back more like bat coronavirus as well. The gaps between the mutation sites are as follows: 69 = 3 × 23; 73; 355 = 3 × 71; 69 = 3 × 23; 44 = 2 × 2 × 11; 67; 35 = 5 × 7; 266 = 2 × 7 × 19; 136 = 2 × 2 × 2 × 17; 155 = 5 × 31. There are three groups of prime numbers involved. The first group is 2,3,5, which belongs to cusps modular (Langlands) prime number. It might be related to the fractal shell-like growing structure. The second group is 7, 11, 19, 23, which belongs to 4k + 3 prime number, also called the Gaussian prime number. The latter is a closed field number on a complex plane, meaning that the numbers form a total ordered chain. It might be attribute to the 3D chain structure of the spike protein. The third group is 31, 67, 71, 73, which belongs to the prime numbers of binary digits. It might be attributed to the long folding structure of the protein.

**Figure 7 entropy-23-00512-f007:**
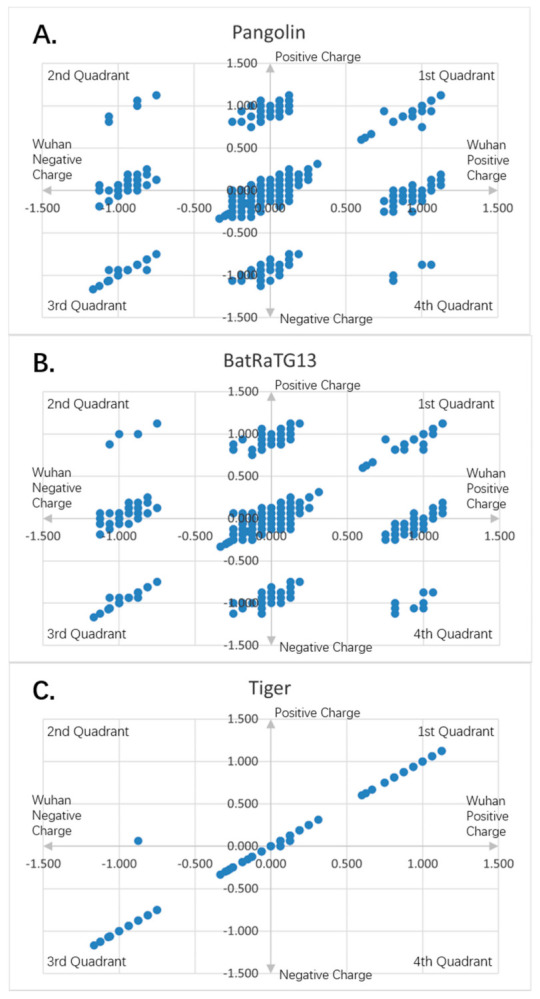
Charge scatter patterns for spike proteins of pangolin, bat and tiger coronaviruses relative to that of Wuhan strain SARS-CoV-2: (**A**) Charge scatter plot of Cartesian coordinates with four quadrants is used to display values for two variables for a set of data. The data are displayed as a collection of points, each having the value of one variable (amino acid charge value from Wuhan strain sequence) to determine the position on the horizontal axis (for Wuhan strain) and the value of the other variable (amino acid charge value from spike protein of pangolin coronavirus) determining the position on the vertical axis (for other). A scatter plot can suggest various kinds of correlations between variables with a certain linear or nonlinear pattern. Correlations may be positive (rising), negative (falling), or neither (uncorrelated). If the pattern of dots slopes from lower left to upper right, it indicates a positive correlation between the variables being studied. If the pattern of dots slopes from upper left to lower right, it indicates a negative correlation. If the dots are continuously connected one after another, it is a simple linear relationship. If the dots form a few islands, it is the nonlinear pattern. The island on the map means there is a domain of related charge event data that is highly correlated within the domain only, but independent from other domains. If both the patterns are there, it is the mixed domain of linear and nonlinear patterns. If within the islands, it is linear, it can be called local linear, globally nonlinear, or piece wised linear. It means only a particular charged piece of the entire sequence is linear correlated within that piece. If the island is viewed as a super dot, and super dots forming a linear relationship, it is called as global linear and local nonlinear. It means the specially charged pieces of the entire sequence are linearly correlated among the pieces. Each piece has its unique electro-biological functions. The first and third quadrants are the pieces where Wuhan strain sequence has the same charge as the pangolin’s. The second and fourth quadrants are the pieces where the Wuhan sequence has the opposite charge to the pangolin viral spike protein. (**B**) Charge scatter pattern of spike protein from bat coronavirus relative to that of Wuhan strain SARS-CoV-2. (**C**) Charge scatter pattern of spike protein from the coronavirus isolated from the New York Zoo tiger relative to that of Wuhan strain SARS-CoV-2. The charge pattern is identical to Figure 8A–C, indicating the tiger virus was transmitted from human SARS-CoV-2 strain.

**Figure 8 entropy-23-00512-f008:**
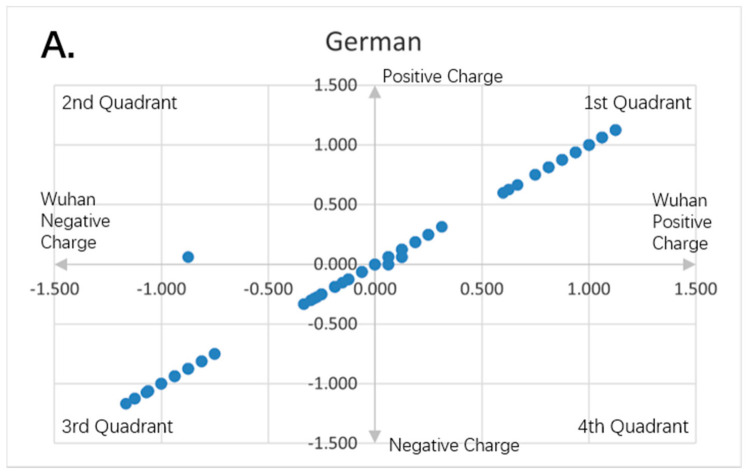
Charge scatter patterns for spike proteins of SARS-CoV-2 strains isolated from various locations relative to that of Wuhan strain SARS-CoV-2. The first and third quadrants of the graphs are the pieces where Wuhan strain sequence has the same charge as the compared sequence. The second and fourth quadrants are the pieces where Wuhan strain sequence has the opposite charge to the compared spike protein sequence. (**A**) Charge scatter pattern of spike protein from SARS-CoV-2 strain isolated in German strain relative to that of Wuhan strain SARS-CoV-2. (**B**) Charge scatter pattern of spike protein from SARS-CoV-2 strain isolated in New York relative to that of Wuhan strain SARS-CoV-2. (**C**) Charge scatter pattern of spike protein from SARS-CoV-2 strain isolated in Beijing wholesale market relative to that of Wuhan strain SARS-CoV-2. (**D**) Charge scatter pattern of spike protein from UK variant B.1.1.7 relative to that of Wuhan strain SARS-CoV-2. There is significant dissimilarity between Wuhan strain SARS-CoV-2 and UK variant B.1.1.7 as compared to panels (**A**–**C**).

**Figure 9 entropy-23-00512-f009:**
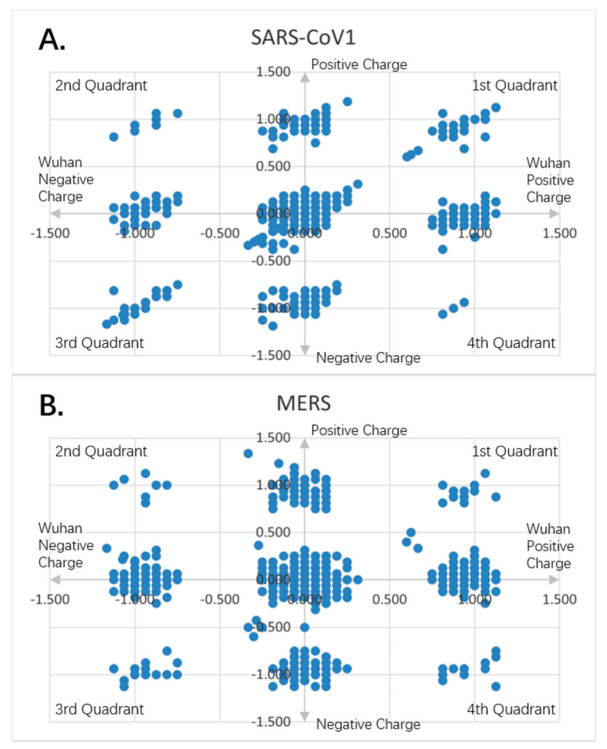
Charge scatter patterns for spike proteins of SARS-CoV-1 and MERS relative to that of Wuhan SARS-CoV-2 strain. The first and third quadrants of the graphs are the pieces where Wuhan strain sequence has the same charge as the compared sequence. The second and fourth quadrants are the pieces where Wuhan strain sequence has the opposite charge to the compared spike protein sequence. (**A**) Charge scatter pattern of spike protein from SARS-CoV-1 relative to that of Wuhan SARS-CoV-2 strain. The pattern shows both linear and nonlinear relationship. (**B**) Charge scatter pattern of spike protein from MERS relative to that of Wuhan SARS-CoV-2 strain. The pattern shows a nonlinear relationship, indicating a strong dissimilarity between MERS and Wuhan strain SARS-CoV-2.

**Figure 10 entropy-23-00512-f010:**
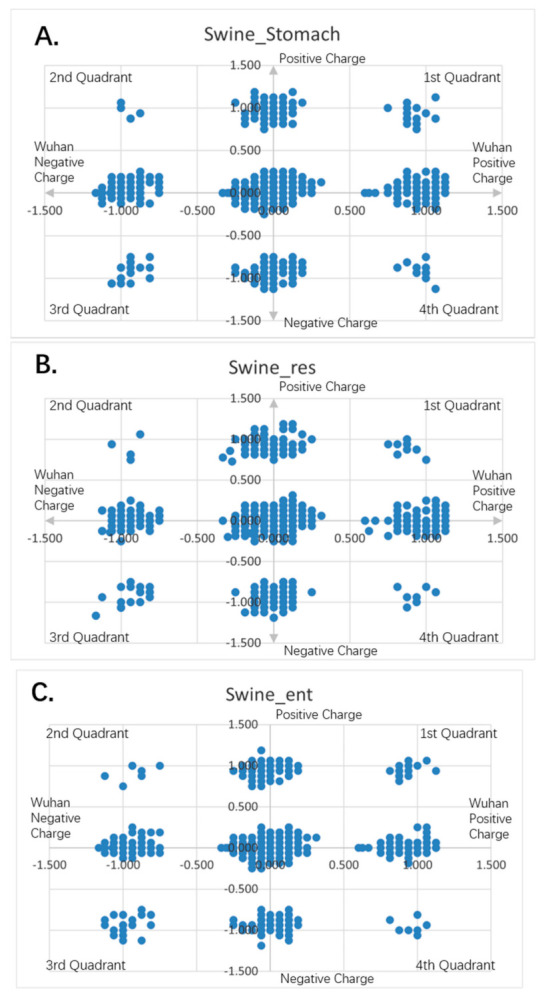
Charge scatter patterns of spike proteins from animal coronaviruses relative to that of Wuhan strain SARS-CoV-2. The first and third quadrants of the graphs are the pieces where Wuhan strain sequence has the same charge as the compared sequence. The second and fourth quadrants are the pieces where Wuhan strain sequence has the opposite charge to the compared spike protein sequence. (**A**) Charge scatter pattern for the coronavirus of swine stomach disease relative to that of Wuhan strain SARS-CoV-2. (**B**) Charge scatter pattern for coronavirus of swine respiratory disease relative to that of Wuhan strain SARS-CoV-2. (**C**) Charge scatter pattern for coronavirus of swine enteritis relative to that of Wuhan strain SARS-CoV-2. (**D**) Charge scatter pattern for feline coronavirus relative to that of Wuhan strain SARS-CoV-2. (**E**) Charge scatter pattern for canine coronavirus relative to that of Wuhan strain SARS-CoV-2. The above patterns show a nonlinear relationship between these viral spike proteins and Wuhan strain spike protein, indicating strong dissimilarity between them.

**Figure 11 entropy-23-00512-f011:**
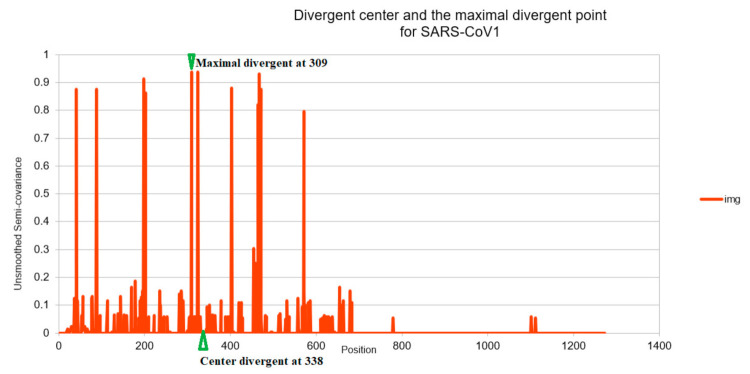
Charge distance illustration for SARS-CoV-1 relative to SARS-CoV-2. The maximal divergent point is defined as the position where the negative semicovariance before been smoothed (we smoothed Figure 1, Figure 2, Figure 3, Figure 4, Figure 5 and Figure 6 for easier comparison) that obtains the peak value. The divergent center is defined as the position where the center of the charge (related to Coulomb force) of the entire sequence landed on (as if all the charges are originated from that one point). The distance (338−309 = 29) between these two positions (after beingbeen normalized) is proportional to the shift distance (18) between SARS-CoV-1 and the SARS-CoV-2 (Wuhan baseline). It is also proportional to the fatality (9.56%) of the virus (SARS-CoV-1) as seen in Table 2.

**Table 1 entropy-23-00512-t001:** Pearson and Semicovariance Coefficient Analysis of Spike Proteins between Wuhan strain SARS-CoV-2 * and other Coronaviruses.

	Pearson Coefficient Analysis													
Pearson Rank	2	4	3	7	6	10	9	5	1	8	1	1	1	1
Source of coronaviruses	Pangolin	SARS-CoV1	Bat RaTG13	Canine	Swine-Stomach	Swine-Res	MERS	Swine-Ent	CoV2-UK (B117)	Feline	NY-Tiger	CoV2-Beijing	CoV2-NY	CoV2-German
Genbank ID/GISAID ID	PCoV_GX-P4L	NC_004718	QHR63300.2	AEQ61968.1	AQT01349	KR270796.1	NC_019843.3	KR061459.1	EPI_ISL_744131	ASB16887.1	MT365033.1	EPI_ISL_46924	QKT21302.1	QJC19431.1
Year of viral isolation	2020	2003	2020	2012	2016	2015	2012	2015	2021	2017	2020	2020	2020	2020
Maximum Pearson Value	0.4911	0.4533	0.4727	0.0810	0.0929	0.0655	0.0684	0.1026	0.9043	0.0748	0.9978	0.9978	0.9978	0.9978
Offset for the above Max #	−6	−18	−4	179	154	−35	74	86	−4	161	0	9	0	0
	Semicovariance coefficient analysis													
Convergent-Covariance Rank	2	4	3	8	7	10	5	6	1	9	1	1	1	1
Convergent Correlation	0.5728	0.5421	0.5693	0.2048	0.2190	0.1813	0.2240	0.2228	0.9192	0.2037	0.9981	0.9981	0.9981	0.9981
Divergent-Covariance Rank	2	3	4	8	5	7	9	6	1	5	1	1	1	1
Divergent Correlation	0.0809	0.0887	0.0967	0.1396	0.1411	0.1271	0.1652	0.1256	0.0145	0.1222	0.0003	0.0003	0.0003	0.0003
Series (conserved region)	900			800	700					600				
Center Convergence (conserved center)	906	905	904	801	784	751	737	727	702	683	658	658	658	658
Center Divergence	318	338	314	691	657	650	698	654	432	635	614	614	614	614
Maximal convergent position	1262	1262	1262	983	983	1262	983	983	1262	278	1262	1262	1262	1262
Maximal divergent position	191	309	558	1107	1107	775	214	49	97	843	614	614	614	614
Convergent-rank	4	10	2	8	9	5	3	7	1	6	1	1	1	1
Number convergent irrelevant positions	76	103	67	81	101	78	75	77	57	79	31	31	31	31
Divergent-rank	3	9	2	8	10	5	4	6	1	7	1	1	1	1
Number divergent irrelevant positions	103	73	106	83	68	98	97	94	109	87	123	123	123	123
Over all rank	2.6	6.0	2.8	7.8	7.4	7.4	6.0	6.0	1.0	7.0	1.0	1.0	1.0	1.0

* GenBank ID for spike protein of Wuhan-Hu-1 SARS-CoV-2: NC_045512.2. The spike protein of Wuhan-Hu-1 virus is used as a reference for the analysis and comparison with other listed coronaviruses in the Table; **#** Offset for the above Max means how much the compared sequence was shifted up or down to obtain the maximum Pearson correlation. For example, the first 6 amino acids of SARS-CoV-2 were cut out from Wuhan strain to line up with pangolin strain sequence to obtain the maximum Pearson so that it is −6. On the other hand, the first 74 amino acids of MERS were cut out to line up with Wuhan SARS-CoV-2 sequence so that it is +74.

**Table 2 entropy-23-00512-t002:** Fatality Rate Related to Sequence Structures’ Pearson Correlation Coefficients.

Coronaviruses	SARS-CoV-1	MERS	SARS-CoV-2
Fatality Rate	9.56% *	34.4% *	2.21% *
Offset Value for the Max Pearson	18	74	9
Correlation to Offset	0.9981		
Coulomb Center to Max Force Ratio	338/309 = 1.094	698/214 = 3.262	614/614 = 1.000
Correlation to Divergence	0.9958		

* Fatality rate obtained from the World Health Organization (WHO) website.

## Data Availability

All the data are available as requested.

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
