# Peer review of "Semicovariance Coefficient Analysis of Spike Proteins from SARS-CoV-2 and Other Coronaviruses for Viral Evolution and Characteristics Associated with Fatality"

_entropy, 2021, doi:10.3390/e23050512_

Round 1
Reviewer 1 Report
The authors compare a fractal-based and a "linear" correlation magnitude by applying both to COVID-19 and various variants. They produce sequences of correlations between the Wuhan and other viruses' spike proteins and point to some regions where the correlations are high. They also feature charge correlation patterns. All results are based on sequences from the data bank.
I do not see what is the point of this work. Since the information is in the exact spike sequence, a simple comparison yields the locations where the proteins are identical or different. As far as I understand the authors consider only the primary protein structure and thus no information on the possible tertiary differences is included in the work. The correlator diagrams are not of high quality while the last group of figures (7-10) have a conspicuously similar structure primarily along the main diagonal while the "island" structure is not explained clearly. Finally, the literature used seems to be for an economics paper and not biophysics.
In all I am not sure what one learns from this work and what are the consequences in biological context. The fact that one may use a fractal correlator that appears to contain a bit more information than Pearson is not sufficient for making this work publishable, unless the authors manage to extract and demonstrate real utility of the results in the viral case.
I cannot recommend the paper for publication in this present form.
Author Response
Please see the attached to reviewers' comments
Reviewer 2 Report
The paper seems of integrest, given global pandemic of SARS-Cov19. The algorithm presented for fatality analysis of the virus uses the novel strategy utilizing semi-covariance coefficient analysis of spike proteins. This is obviously of interest for the global pharmacological industry. Although the work focuses on a new method, it is adviced to allow non-experts to familarize with the proposed methodology. Therefore, a step-by-step intorduction to the new method will substantially improve the impact of the paper.
Specific issues:
a) In the intorduction, the second paragraph should be rewritten to the extent to allow understanding of the link of FDH with the proposed methodology? Why is it mentioned? Can it be clarified?
b) In the methods section 2.2 the theoretical derivation of this methodology should be simplified to allow a non expert reader to understand the major difference between the pearson coefficient and the proposed semi-covariance coeffcient methodology. This is also improtant to highlight the terminology from Table 1.
c) it is not clear how the presented methodology translates into an Excel sheet. If the Authors claim it can be used there, there should be a clear instruction on how this can be done.
d) The Authors are encouraged to add information on how they "... calculated the sequences of the spike proteins [18]..."
The paper seems to be already published in preprints, and this should be carefully agreed with the journal policy.
Author Response

(The authors gave the same response as above.)

Reviewer 3 Report
I have the following questions for the authors:
1) The semi-covariance coefficient and the non-linear correlation relationships were calculated using moving averages of values above and below the point (2 in each direction for the semi-covariance coefficient and 16 in each direction for the non-linear correlation). Can I assume that the number of values included in the moving averages was tested to find optimum sizes? Would the authors please comment on that, especially on the effect of altering the sizes and how the optimum was determined?
2) In figures 1 through 3 the position values (x axis) are superimposed with the negative portions of the curves. These numbers are frequently obscured. It would be helpful for the authors to move the position values to the bottom of the graph (at least below the negative curves) for easier reading.
3) I think it would be helpful if the authors could include a figure which illustrates the definition of "distance between the divergent center and the maximal divergent point" so that more readers will better understand this important measurement.
Author Response

(The authors gave the same response as above.)

Reviewer 4 Report
Dear Editor and Authors,
the last 3 decades the analysis of data with tools of the complexity theory is a very interesting area. When the system reaches in critical states, the parameters of the system change, showing the dynamic phase transition. The complexity tools describe very fine the projection of the phase space of the dynamical system and the structure which makes the dynamical orbit of the system with a non-linear way forming a complex system far from equilibrium. The proximity of the systems far from equilibrium in critical states make dynamic like low dimensional chaos, soc, long range correlations, strange kinetics, islands, etc.
Many researchers used complexity tools to analyze the DNA raw data investigated the fundamental dynamics of the DNA system and the connection with the biological effects like autoimmune diseases, cancer, mutations, fatality etc.
In this interesting study the identification of the complex character based in the semi-covariance coefficient analysis of the spike proteins from SARS-CoV-2 and other coronaviruses are presented. The Fractal DNA hypothesis corresponds to similarities or correlations (positive or negative)) reveals trends and patterns when the DNA system change behavior.
The authors presented a clear and efficient methodology to apply the semi-covariance coefficient analysis in the spike proteins data from a team of coronaviruses included of course the SARS-CoV-2 due to pandemic conditions. Finally, the study connects trends and patterns for the mutations and the fatality of this virus.
The sound methodology and rigorous mathematical analysis carried out within this work. The presentation of the theoretical framework and the results of analysis are very understandable. The approach of the paper is in a right way to understand the complexity and the criticality behavior of the DNA system.
This study has a very well-written English text and a very good presentation.
Finally, I suggest publication of the paper in its present form.
In the future I will suggest to the authors to compare the results of this study with machine learning approaches for classification and prediction to categorized the correlations (positive or negative) of the spike proteins data based on of the above methodology.
Best Regards
Reviewer
Author Response

(The authors gave the same response as above.)

Round 2
Reviewer 1 Report
The revised version is improved and the authors made changes in the direction of making their points clearrer. I recommend the paper for publication.
Reviewer 2 Report
The Authors have addressed all the comments. Hence, the Referee can recommend the work for publication.
This manuscript is a resubmission of an earlier submission. The following is a list of the peer review reports and author responses from that submission.